# Development and Optimisation of HILIC-LC-MS Method for Determination of Carbohydrates in Fermentation Samples

**DOI:** 10.3390/molecules26123669

**Published:** 2021-06-16

**Authors:** Dmitri Pismennõi, Vassili Kiritsenko, Jaroslav Marhivka, Mary-Liis Kütt, Raivo Vilu

**Affiliations:** 1Center of Food and Fermentation Technologies (TFTAK), Akadeemia tee 15A, 12618 Tallinn, Estonia; vassili@tftak.eu (V.K.); jaroslav@tftak.eu (J.M.); maryliis@tftak.eu (M.-L.K.); raivo@tftak.eu (R.V.); 2Department of Chemistry and Biotechnology, Tallinn University of Technology, Akadeemia tee 15, 12618 Tallinn, Estonia

**Keywords:** saccharides, HILIC-LC-MS, *Streptococcus thermophilus*, fermentation

## Abstract

Saccharides are the most common carbon source for *Streptococcus thermophilus*, which is a widely used bacterium in the production of fermented dairy products. The performance of the strain is influenced by the consumption of different saccharides during fermentation. Therefore, a precise measurement of the concentrations of saccharides in the fermentation media is essential. An 18-min long method with limits of quantitation in the range of 0.159–0.704 mg/L and with ^13^C labelled internal standards employing hydrophilic interaction chromatography coupled to mass spectrometric detection-(HILIC-LC-MS) allowed for simultaneous quantification of five saccharides: fructose, glucose, galactose, sucrose, and lactose in the fermentation samples. The method included a four-step sample preparation protocol, which could be easily applied to high-throughput analysis. The developed method was validated and applied to the fermentation samples produced by *Streptococcus thermophilus*.

## 1. Introduction

*Streptococcus thermophilus* is a gram-positive facultative anaerobic bacterium mostly known for its role in the production of fermented dairy products. As a part of the lactic acid bacteria (LAB) group, it is a widely studied and well-known microorganism. As production of lactic acid by *S. thermophilus* is dependent on carbohydrate utilisation, the choice of available carbohydrate is usually dictated by strain ability to digest certain disaccharides—mainly sucrose and lactose [1]. Bacterial consumption of monosaccharides—fructose, glucose. and galactose, was found to be suppressed in most common strains. Nevertheless, the mutant strains, able to consume monosaccharides, were created to study the alternative ways for lactic acid production by utilisation of low-molecular carbohydrates [2,3,4].

Carbohydrates are a vast class of chemical compounds with a similar structure comprised of either furanose or pyranose skeleton core. Carbohydrate analysis employing chromatography has a long history starting from paper partition chromatography of selected monosaccharides in 1949 [5]. The advances in the chromatographic field have helped to achieve better separation and selectivity [6,7]. The main path to analyse carbohydrates was to use either gas chromatography with derivatisation of saccharides or liquid chromatography employing ion-exchange resins [8]. The development of novel stationary phases for both gas and liquid chromatography increased the number of applications where carbohydrate analysis could be performed from samples obtained from various sources, i.e., raw nutritional materials, animals, bacteria, humans, and so forth [9,10,11,12]. At the same time, developments in ion chromatography and electrophoresis allowed to measure carbohydrates in similar matrices creating the alternative ways for the measurements of saccharides [13,14,15,16]. Nowadays, the most commonly used method is to measure carbohydrates to use ion-exchange resin with a refractive index (RI) detector, as carbohydrates do not have any chromophores. The use of this combination is shown in Association of Official Analytical Chemists (AOAC) or International Organization for Standardization (ISO) methods [17]. The main disadvantage with RI-based detection is a relatively low selectivity, sensitivity, and elution program restriction compared to more advanced detection techniques such as evaporative light scattering detector (ELSD) [18,19], charged aerosol detection (CAD) [20] or mass spectrometric (MS) detector [21,22,23]. Out of advanced methods, MS-based detection offers the most selectivity as active mass filtration could clean up a substantial portion of noise interference originated from the sample matrix [24]. Therefore, the liquid chromatography coupled to mass spectrometer (LC-MS) system became a prominent choice to perform carbohydrate analysis with little to no extensive sample preparation [25,26]. 

The aim of this work was to develop and validate a rapid and sensitive method for the quantitative determination of five saccharides: fructose, glucose, galactose, sucrose, and lactose, by employing a rapid, selective, and sensitive methodology based on hydrophilic interaction chromatography coupled to mass spectrometric detection (HILIC-LC-MS) and isotopically labelled glucose and lactose as internal standards. The method development included optimisation of sample preparation, validation and application of the method towards the determination of carbohydrates metabolised by *S. thermophilus* in fermentation broth samples.

## 2. Results and Discussion

### 2.1. Chromatographic and Mass Spectrometric Optimisation

The initial screening involved testing of the performance of several columns with hydrophilic interaction (HILIC) stationary phase. The testing of Waters BEH HILIC and BEH Amide revealed that even though the columns are clearly capable of separation between mono- and disaccharides, the inter-class separation of closely matched carbohydrates is impossible (Figure 1).

During method scouting, it became evident that BEH HILIC column could not achieve an acceptable separation of 3 monosaccharides of interest. Employing Waters BEH Amide column showed the separation between fructose and glucose-galactose pair close to the baseline. The separation of the glucose-galactose pair requires more resolving power as the epimer separation is proven to be complex. The acceptable separation between epimers was reached using Phenomenex Luna Omega Sugar column, which allowed a repeatable and precise determination of closely eluting glucose and galactose (Figure 2). 

To improve the sensitivity of measured analytes, the decision to enrich the mobile phases with guanidine hydrochloride solution was made to facilitate the formation of [M + Cl]^−^ adduct instead of [M-H]^-^ ion. The addition of chloride ion to saccharide molecules provided better ionisation, cleaner spectra at the baseline level (Appendix A). It thus decreased the amount of sample injected on the column to achieve a satisfactory chromatographic and mass spectrometric result [27,28]. The variations in flow rate were also studied as the mock-up method was transferred from the column with a larger inner diameter, making it incompatible with the current column choice. Therefore, several different flow rates were evaluated to achieve optimal separation between all analytes: 300, 313 and 350 µL/min (Appendix A). The flow rate 313 µL/min resulted in the most optimal separation among targeted carbohydrates. The column temperature was also studied, and two temperatures were tested: 25 and 35 °C. The higher temperature was ruled out as unfavoured due to more unsatisfactory performance in terms of chromatographic separations, which was in accordance with reports in the literature [29].

### 2.2. Sample Preparation Optimisation and Measurement of Fermentation Samples

Several variants of sample preparation were proposed during initial consideration for adequate sample preparation for maximum elimination of matrix components (Table 1). 

As the protocols were being tested firstly with a simulated matrix composed of chemically defined medium (CDM) [30] and external standards, it was found that variants 2, 3, 4 and 6 did not produce expected results as analytes were detected in lower amount compared to other protocols or no analytes of interest were found during the measurement (Appendix A). Protocol nr 1 and 5 were chosen for additional evaluation as their performance with simulated matrix was found to be acceptable. The optimised protocols nr. 1 and 5 were applied towards harvested fermentation broth. It was found that protocol nr 1 produced a higher number of impurities which negatively affected MS performance by leaving more residue on the source cone compared to protocol nr 5, which included the usage of molecular weight cut-off filters (Figure 3).

After sample preparation protocol optimisation was completed and sample preparation protocol nr 5 was chosen as the primary option to perform sample preparation. Six samples obtained during the fermentation process at different time points were analysed for carbohydrate content (Table 2). 

The samples were injected in triplicates. (Figure 4).

It was found that neither of the samples contained fructose at levels exceeding the limit of quantification. Other samples contained either sucrose or lactose at a higher concentration due to its presence in CDM. It was shown in samples B03F2_1 and B3F2_2 that the bacterium was able to produce monosaccharides by breaking down larger saccharide. In other samples, no by-products of disaccharide breakdown were detected, which could indicate that bacteria would utilise either sucrose or lactose as a carbon source for the production of other molecules such as organic acids. The results were subjected to carbon balance calculations based solely on the saccharide content in the fermented and blank samples. The calculations showed that measured saccharide content is correlatable with calculations e.g., carbon balance calculation errors were 25.6 ± 16.1% (n = 6) on average between fermented samples. The obtained values were found to be acceptable as calculations did not take into the account presence of other organic materials commonly present in the fermentation broth.

### 2.3. Validation Results

When the development and optimisation of methodology were finished, validation was performed to evaluate the method linear range, limits of detection and quantifications, recoveries and the stability of prepared samples. First of all, linear range and linearity were evaluated via the repeated measurements of standard solutions consisting of 8 individual points obtained from stock’s serial dilution (Table 3).

After linearity was found to be acceptable (R2 ≥ 0.99 or higher) for all sugars in this study, the repeatability of the method was studied. Repeatability of retention times and peak areas were studied first with six replicate injections of standard solution (Table 4). The repeatability of the method was studied across four independent days to confirm the stability of the retentions time and peak areas of the analytes.

Recovery of the sample preparation was determined by analysing fermentation samples, whereas ^13^C-internal standards were spiked prior to sample preparation steps and compared to sample preparation procedure where ^13^C-internal standards were added in the last stage of the procedure. Recovery was calculated according to Equation (1): Recovery = Peak area in a spiked sample/Peak area in a non-spiked sample ∗ 100(1)

The recovery experiments consisted of injections of 2 samples in triplicate. The recorded recovery values were 103.73 ± 1.69% and 111.04 ± 2.80% for ^13^C-labeled glucose and ^13^C-labeled lactose, respectively. The obtained values are in the ±20% range. Furthermore, we have investigated the stability of the prepared standard solutions in a ready-to-use form stored at +4 and −20 °C. The prepared standards were stable at +4 degrees for one week whereas peak area of analytes has not changed by more than 5%. The analytes stored at −20 degrees for 1 month showed stable retention factors or all measured compound except for sucrose, which response factor after 1 month of storage at −20 °C had changed by 15%.

## 3. Materials and Methods

### 3.1. Reagents and Chemicals

Standards of mono- and disaccharides: d-fructose, d-glucose, d-galactose, d-(+)-sucrose, d-lactose monohydrate, and ammonia solution (25%, LC-MS LiChropur™ grade) were obtained from Sigma-Aldrich (Darmstadt, Germany). Glucose-^13^C_6_ (Glu-^13^C_6_, U-^13^C_6_, 99%, chemical purity 98%) and lactose monohydrate (Lac-^13^C_6_, UL-^13^C_6_glc, 98%+) were procured from Cambridge Isotope Laboratories Inc. (Tewksbury, MA, USA). Ultrapure water (18.2 MΩ.cm) was prepared with MilliQ^®^ Direct-Q^®^ UV (Merck KGaA, Darmstadt, Germany). Acetonitrile (MeCN; LiChrosolv, HPLC gradient grade), and guanidine hydrochloride (GuHCl; ≥99%) were acquired from Sigma-Aldrich (Darmstadt, Germany). Biotage Isolute® PLD+, C18 and NH_2_ were procured from Biotage Sweden AB (Uppsala, Sweden). Amicon Ultra-0.5 centrifugal filter unit (3 kDa) and Millex-LCR filters (Pore size 0.2 µm, Filter Dimension 13 mm) were obtained from Merck KGaA (Darmstadt, Germany) and Microsep Advance Centrifugal Devices with Omega Membrane 1K filter unit was purchased from Pall Corporation (Port Washington, NY, USA).

### 3.2. Preparation of Standard Solutions

The stock solution of each individual saccharide was prepared in MilliQ^®^ water and stored at −20 °C. Solutions of isotopically labelled standards were dissolved in aqueous MeCN (50%, *v/v*) and stored at −50 °C. Working solutions for the determination of analytes were prepared firstly by diluting the stock solution with 100% MeCN, and after each working solution was prepared in aqueous MeCN (50%, *v/v*) water. Calibration curves were built for fructose (0.39–49.875 ppm), glucose (0.506–64.800 ppm), galactose (0.388–49.600 ppm), sucrose (0.467–59.750 ppm) and lactose (0.384–49.100 ppm). Glucose-^13^C_6_ and lactose-^13^C_6_ were added prior to injection to the autosampler vial, and their concentration in the vial was set at 15.925 and 12.825 ppm, respectively. The calculations of calibration curves used response factors, which were calculated according to Equation (2).
Response Factor (RF) = Area of analytes × (Concentration of internal standard/Area of internal standard)(2)

Calibration curves were built using eight-point measurements of serially diluted standards. The regression was found by fitting a point to a linear equation.

### 3.3. Liquid Chromatography

Samples were analysed using a Waters UPLC^®^ system (Waters Corporation, Milford, MA, USA) coupled with a Waters Quattro Premier XE Mass Spectrometer equipped with ZSpray™ Source and controlled by Waters MassLynx™ 4.1 (V4.1 SCN805, Waters Corporation, Milford, MA). Mobile phases were as follows: (A) 99% MilliQ^®^ + 1% MeCN + 1 mg/L of GuHCl and (B) 99% MeCN + 1% MilliQ^®^ + 1 mg/L of GuHCl. Weak needle wash was composed of 10% MilliQ^®^ in MeCN (*v/v*), and strong wash needle consisted of 10% MeCN in MilliQ^®^ (*v/v*). Seal wash was aqueous MeCN (50%, *v/v*). Samples were stored at an autosampler which held temperature at 8 °C. The injection volume was 2 µL. Several columns were tested: Waters Acquity UPLC^®^ BEH HILIC (2.1 × 100 mm, 1.7 µm, Waters Corporation, Milford, MA), Waters XBridge^®^ BEH Amide XP (3.0 × 150 mm, 2.5 µm, Waters Corporation, Milford, MA), Phenomenex Luna Omega Sugar column (2.1 × 150 mm, 100 Å, 3 µm, Phenomenex Inc., Torrance, CA, USA). To prevent harm to any analytical column, ACQUITY UPLC Column in-line filter unit (Waters Corporation, Milford, MA) with installed 0.2 µm stainless steel filter was used in all experiments with all tested columns. The column temperature was held at 25 degrees of Celsius for the duration of all experiments. The gradient was as follows: 0–10 min linear gradient 10–25% A, 10–12 min hold at 25% A, 12.01–14 min hold at 35% A, 14.01–18 min hold at 10% A. Flow rate was set at 313 µL/min.

### 3.4. Mass Spectrometry

The analytes were ionised under negative electrospray ionisation conditions with optimised source conditions. The source temperature was set at 120 °C, high-purity nitrogen was fed into the source at 50 L/h (cone) and 800 L/h (desolvation) and heated to 350 °C. The capillary voltage was set at −2.3 kV, cone voltage at 25 V and extractor voltage 3 V. The values for efficient ionisation were found by infusing a standard solution of individual saccharide (*ca* 25 ppm) in 50% aqueous MeCN (*v/v*) at the combined flow from UPLC and integrated syringe pump at 250 µL/min. For measurement of analytes, single ion monitoring (SIR) experiments were chosen as saccharides possess no valuable fragments for ubiquitous identification, therefore, making it unnecessary to perform multiple reaction monitoring (MRM) type of experiments. The SIR channels were based on molecular ion with added chloride as [M+Cl]^-^. Data acquisition was performed in Waters MassLynx™ V4.1 (SCN805, Waters Corporation, Milford, MA, USA). Data analysis was performed in Waters QuanLynx™ V4.1 (SCN805, Waters Corporation, Milford, MA, USA) and Microsoft Excel (Microsoft 365 Apps for enterprise).

### 3.5. Bioreactor Experiments, Bacteria Growth Description

*Streptococcus thermophilus* was inoculated into fresh M17 medium (lactose as carbon source) at the rate of 1e^7^ cfu/mL (1%) and cultivated until OD_600_ reached a value of 0.8. Then the culture was inoculated into a bioreactor containing 300 mL of CDM achieving 100-fold dilution. CDM contained either sucrose or lactose as a carbon source. Cultivation was conducted in anaerobic conditions maintained by the constant flow of N_2_ into the medium flask and N_2_:CO_2_ mixture (80:20, *v/v*) into the reactor vessel at 150 and 300 mL/min, respectively.

Culture outgrowth was monitored by the rate of medium acidification and using the turbidimetric sensor. After 7 h of batch growth, the stability of culture was achieved, and flowthrough was initiated with a dilution rate of 0.25 h^−1^. The flow was maintained for 20 h, ensuring culture stabilisation. At the chemostat point, the culture samples were taken for subsequent analysis. HPLC samples were centrifuged at 14,000 rpm for 10 min. Supernatants were frozen and stored at −20 °C.

### 3.6. Sample Preparation

Frozen samples were fully thawed at room temperature until a clear solution was obtained. Thawed samples were serially diluted 100-fold before further steps. Diluted samples (1000 µL) were firstly centrifuged at 14,000 rpm for 10 min to remove any remaining solid residue. The supernatant (500 µL) was then transferred to a 3 kDa molecular weight cut-off (MWCO) filter (Amicon^®^ Ultra-0.5, Merck KGaA, Germany). The MWCO filter was then centrifuged at 14,000 rpm for 20 min. The supernatant obtained was diluted with a 50% aqueous MeCN (*v/v*) mixture containing Glu-^13^C_6_ and Lac-^13^C_6_ 2-fold before analysis. 

### 3.7. Method Validation

The developed method was assessed for linearity (as a correlation coefficient of R^2^ of calibration curve), the limit of detection and quantification (as the standard deviation of the measured sample at the lowest calibration points multiplied by 3 or 10, respectively), recovery (as spiked sample vs. un-spiked) and matrix effect [31]. 

## 4. Conclusions

In summary, we have developed the HILIC-LC-MS method for the rapid and simultaneous determination of five saccharides in just 18 min without employing complex sample preparations steps. The methodology can be applied to the simplest instrumentation consisting of liquid chromatograph and single quadrupole mass detector. The addition of a common mobile phase additive such as guanidine hydrochloride is a viable option to increase signal insensitivities while reducing the baseline noise. The mass spectrometric detection helps with the selectivity of the methodology as mass spectrometer could filter out matrix interfering components thus providing cleaner and unambiguous spectra. The simultaneous measurement of fructose, glucose, galactose, sucrose, and lactose could be done in a fraction of time and less consumed solvent compared to classical methods used elsewhere [10,11,16,32]. The method developed here suits for both to identify and characterise the metabolism of various starter cultures (like here with *Streptococcus thermophilus*) as well as to detect the sugar profile of different food matrices. Furthermore, the monosaccharide quantities could be used in a calculation of the carbon balance in a single cell model (SCM) to assess the productivity of the strain during fermentation. The applicability of the method for quantification of the larger oligosaccharide chains together with smaller saccharides, for example, to analyse their consumption by gut microbiota, could be further determined in the future.

## Figures and Tables

**Figure 1 molecules-26-03669-f001:**
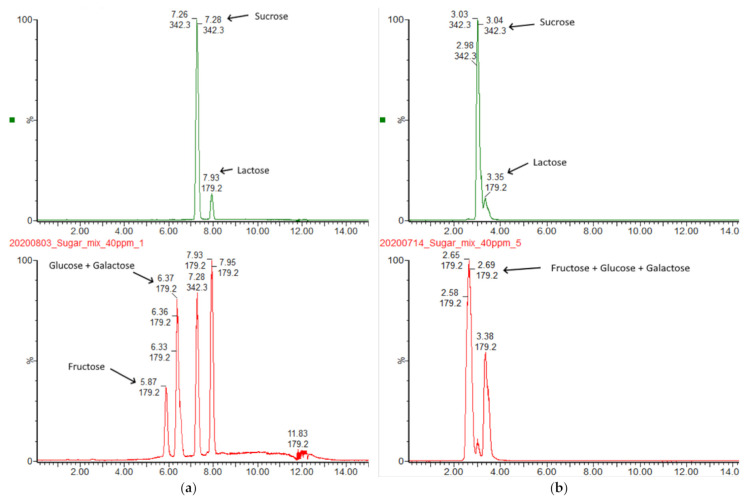
(**a**) Separation of mono- and disaccharides on Waters BEH Amide column with mobile phases containing 0.1% NH_4_OH; The elution was achieved using gradient elution at 170 µL/min flow rate. Solvents were: A—MilliQ + 0.1% NH_4_OH, B—MeCN + 0.1% NH_4_OH. Gradient program was: 0.0–10 min linear ramp 0.1–60% A, 10.01–25.00 hold at 0.1% A. The detection was performed in Single-Ion-Reaction (SIR) mode (**b**) Separation of mono- and disaccharides on Waters BEH HILIC column with mobile phases containing neat solvents. The elution was achieved using gradient elution at 170 µL/min flow rate. Solvents were: A—MilliQ, B—MeCN. Gradient program was: 0.0–20 min linear ramp 0.1–40% A, 20.01–30.00 hold at 0.1% A. The detection was performed in Single-Ion-Reaction (SIR) mode at a concentration level of 10 µg/mL for all saccharides.

**Figure 2 molecules-26-03669-f002:**
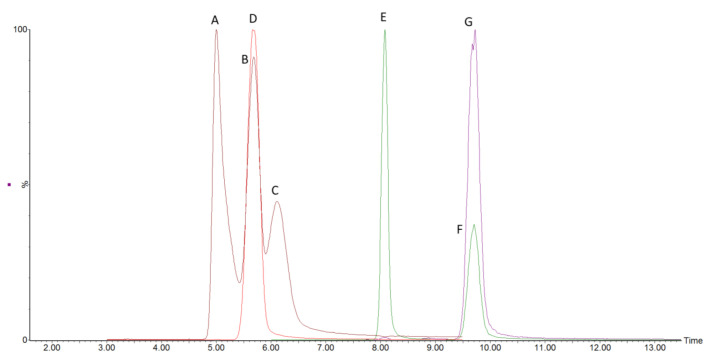
The representative chromatogram of separated mono- and disaccharides with internal ^13^C-labeled standards on the Luna Omega Sugar column. Peaks are labelled as: A—fructose, B—glucose, C—galactose, D—glucose-^13^C_6_, E—sucrose, F—lactose and G—lactose-^13^C_6_. Peak heights are normalised.

**Figure 3 molecules-26-03669-f003:**
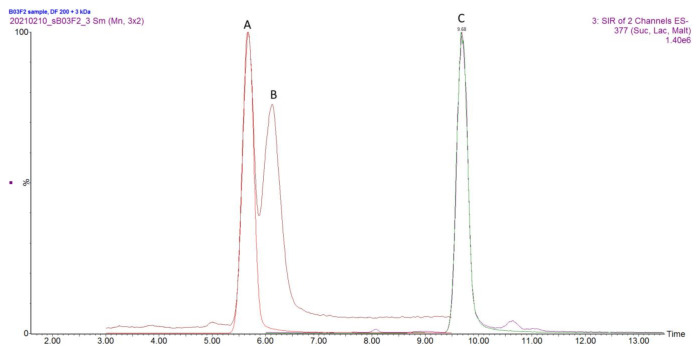
The overlaid chromatogram of the fermented sample subjected to extraction protocol nr. 5. Peaks are labelled as follows: A—Glucose and Glucose-^13^C_6_, B—Galactose, C—Lactose and Lactose-^13^C_6_.

**Figure 4 molecules-26-03669-f004:**
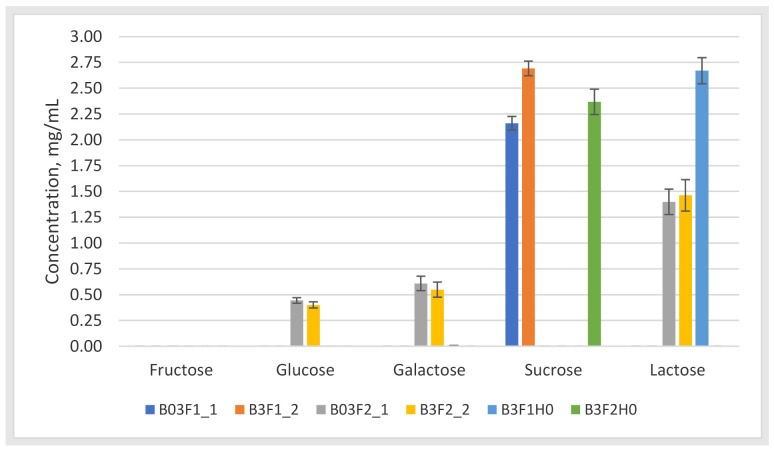
Quantified carbohydrates (mg/mL) in the measured samples. The bars represent the concentration levels of the found saccharides in the fermentation samples. The sample naming follows B stands for batch fermentation; number 3 or 03 states the number of experiments, F1 or F2 states the reactor number and _1 or_2 states the number of parallel. H0 denotes 0 h sampling point.

**Table 1 molecules-26-03669-t001:** Comparison of sample preparation protocols.

Protocol Nr.	1	2	3	4	5	6
Step 1	Dilute 100 times with MilliQ					Dilute 50 times with MilliQ
Step 2	Centrifuge at 14000 rpm for 10 min					
Step 3	Filter through 0.2 µm filter	Dilute 2 times with MeCN: MilliQ mixture containing ^13^C ISTD		Pass through 1 kDa MWCO filter	Pass through 3 kDa MWCO filter	Dilute 4 times with MeCN: MilliQ mixture containing ^13^C ISTD
Step 4	Dilute 2 times with MeCN: MilliQ mixture containing ^13^C ISTD	Pass through Isolute PLD+ cartridge	Pass through Isolute NH_2_ cartridge	Dilute 2 times with MeCN: MilliQ mixture containing C13 ISTD		Pass through Isolute NH_2_ cartridge

**Table 2 molecules-26-03669-t002:** Description of the harvested fermentation samples produced by *S. thermophilus.*

Sample Name	Sampling Time Point, h	Carbohydrate Source
BO3F1_1	24	Sucrose
B3F1_2	24	Sucrose
BO3F2_1	24	Lactose
B3F2_2	24	Lactose
B3F1H0	0	Lactose
B3F2H0	0	Sucrose

**Table 3 molecules-26-03669-t003:** The linear range, regression equation, limits of detection. and quantification of five saccharides.

Analyte	Linear Range, µg/ml	Linear Regression	R^2^	LoD ^1^, mg/L	LoQ ^2^, mg/L
Fructose	0.77–49.88	y = 1.3611x + 0.9873	0.9974	0.189	0.629
Glucose	0.51–64.80	y = 0.6921x + 0.0765	0.9993	0.080	0.268
Galactose	0.39–49.60	y = 0.3764x − 0.0112	0.9958	0.067	0.220
Sucrose	0.93–59.75	y = 1.2610x + 0.9776	0.9935	0.232	0.704
Lactose	0.38–49.10	y = 1.0851x − 0.0076	0.9996	0.048	0.159

^1^ LoD = Blank mean value + 3.3*standard deviation at LLOQ; ^2^ LoQ = Blank mean value + 10* standard deviation at LLOQ.

**Table 4 molecules-26-03669-t004:** Repeatability of retention times and peak areas of measured carbohydrates.

Analyte	Mass of Measured Ion, m/z	Retention Time, min	Retention Time RSD, %		Peak Area RSD %	
			Inter-Day, % (n = 6)	Intra-Day, % (n = 4)	Inter-Day, % (n = 6),	Intra-Day, % (n = 4)
Fructose	215	5.06	0.26	1.27	2.90	3.75
Glucose	215	5.75	0.42	1.59	2.69	4.01
Galactose	215	6.18	0.50	1.50	3.62	3.28
Sucrose	377	8.16	0.23	1.32	3.46	2.96
Lactose	377	9.81	0.19	1.60	4.70	2.40
Glucose—^13^C_6_	221	5.70	0.39	0.59	2.13	1.48
Lactose—^13^C_6_	383	9.73	0.44	0.50	4.63	2.23

## Data Availability

Data is contained within the article.

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
