# Peer review of "Development and Optimisation of HILIC-LC-MS Method for Determination of Carbohydrates in Fermentation Samples"

_molecules, 2021, doi:10.3390/molecules26123669_

Round 1
Reviewer 1 Report
The manuscript by Pismennoi, et al. shows a HILIC-LC-MS method to quantify some monosaccharides and disaccharides in fermentations by Streptococcus thermophilus. The authors have validated the method well and this method has some advantages such as low LOD and ok run time. The authors should provide greater detail in their literature review and provide more chromatograms to better support their work. Therefore, I recommend acceptance after major editorial revisions.
The authors mention LC and GC based methods for monitoring carbohydrates, however, they do not discuss the many other carbohydrate monitoring methods proposed in the literature. For example, ion chromatography and capillary electrophoresis, here is one such reference with both, Simple and robust monitoring of ethanol fermentations by capillary electrophoresis Oliver J.D., Sutton A.T., Karu N., Phillips M., Markham J., Peiris P., Hilder E.F., Castignolles P. (2015) Biotechnology and Applied Biochemistry, 62 (3) , pp. 329-342.
The advantages and disadvantages of this HILIC method should be discussed in comparison to other reported fermentation monitoring methods in the literature. This method is said to have a simple preparation however there are 6 preparation steps and the fermentation monitored does appear as complex as other ethanol fermentations of substrates such as lignocelluloses. To claim the simple sample preparation, rapid detection, simple instrumentation and improved sensitivity direct literature comparisons are needed. For this paper I only believe there is improved sensitivity and slightly faster analysis times.
Other comments:
There is no chromatogram of an injection of a fermentation mixture, this must be added.
Figures 1 and 2 require further details such as the mobile phase and gradient parameters used, what analytes were injected and at what concentrations as well as if it is single or total ion mode.
Lines 82-95 the effect of additives, temperature and flow rate are discussed, figures showing these results should be provided in supporting information.
Lines 101-111 representative chromatograms demonstrating the advantages in protocols should be added to the supporting information to assist the reader understand these optimizations.
Many abbreviations are given without their full name at first appearance, eg: HILIC, AOAC, ISO, CDM and others. Please provide the full names when they first appear in the text.
Table 3 Please add how LOD and LOQ were defined as a footnote or into the caption.
Table 4 There is inter-day and intra-day repeatability of retention time. Can you provide inter and intra day RSDs for the peak area?
Lines 150-151 What is meant by stable? What was the percentage change over time?
Lines 141-148 I think it should be made clear that this is the recovery of the sample preparation and not the recovery of the whole process including after the column. Additionally, the way Equation 1 is written does not match the text. I believe it should be = area of sample spiked in 1st step/last step x 100
Please provide more details about what is in CDM as it is important to understand the complexity of the matrix.
Reviewer 2 Report
In my opinion, the development and validation of a method to determine saccharides is not a suitable topic for Molecules.
Reviewer 3 Report
The manuscript titled “Development and optimisation of HILIC-LC-MS method for determination of carbohydrates in fermentation samples” presented by Dmitri Pismennõi and co-workers have reported an interesting work about the determination of carbohydrates in fermentation samples. However, the quality of the manuscript could be increased, and some points need to be clarified before the publication. • Line 13: Please, use full name before the acronym. • Line 13-15: Please, rephrase. This sentence is not well written. • Line 27, Line 30: The references should be added before the end of the sentences, and the dot should be added after references. Please, correct and standardize throughout the text. • Elements of scientific novelty about the developed method should be deeply presented in the last paragraph of the Introduction. • Line 82: Please, avoid using personal form in scientific paper, as “we decided”. Please, rephase these (for example: “the mobile phase composition was modified…” etc.). • Table 1: Does MQ and MilliQ means both MilliQ-water? Please, always use the same abbreviation in the manuscript. • Line 130: linearity range and method linearity are the same thing. Please, correct. • Due to the fact that I noticed some grammar mistakes, I think authors should revise the language.Author Response
Please see the attachment

Reviewer 4 Report
Dear authors
Attached the comments related to the manuscript.
Best regards

Round 2
Reviewer 1 Report
The manuscript by Pismennoi et al. has been significantly improved and would be acceptable for publication after minor corrections.
I am not convinced that this method is simpler and faster than several other fermentation monitoring methods. Instead of simply saying ‘rapid’, ‘sensitive’ and ‘simple’ the authors should state the analysis time, LOD and sample preparation steps. In particular this should be in the abstract and conclusion.
In Figure 3 why do some traces start at 6 min and others finish at 9.5 min? It is suspicious that the full chromatogram from 0 min to 13 min is not shown. Furthermore, the traces are not labelled to inform the reader what each trace is.
Figures S1-S3 the peaks should be labeled and the difference between the red and green traces stated.
Reviewer 4 Report
The authors have addressed the major issues.
Author Response
Dear reviewer,
Thank you for providing valuable feedback in the previous round of review.